# RXR Agonist V-125 Induces Distinct Transcriptional and Immunomodulatory Programs in Mammary Tumors of MMTV-Neu Mice Compared to Bexarotene

**DOI:** 10.3390/biomedicines14010080

**Published:** 2025-12-30

**Authors:** Afrin Sultana Chowdhury, Lyndsey A. Reich, Karen T. Liby, Elizabeth S. Yeh, Ana S. Leal

**Affiliations:** 1Department of Biochemistry, Molecular Biology, and Pharmacology, Indiana University School of Medicine, Indianapolis, IN 46202, USA; afschow@iu.edu (A.S.C.); esyeh@iu.edu (E.S.Y.); 2Indiana University Melvin and Bren Simon Comprehensive Cancer Center, Indianapolis, IN 46202, USA; ktliby@iu.edu; 3Department of Pharmacology and Toxicology, Michigan State University, East Lansing, MI 48824, USA; lyndsey.reich@sutterhealth.org; 4Department of Medicine, Indiana University School of Medicine, Indianapolis, IN 46202, USA

**Keywords:** retinoid X receptor (RXR) agonist, HER2^+^ breast cancer, tumor microenvironment

## Abstract

**Background:** The retinoid X receptor (RXR) is a ligand-activated nuclear receptor that heterodimerizes with numerous partners to regulate diverse transcriptional programs. RXR agonists, including the FDA-approved drug bexarotene, show anti-tumor activity but are limited by adverse side effects. V-125 is a next-generation RXR agonist engineered for improved selectivity, pharmacokinetics, and reduced lipogenic effects. This study compares the molecular and functional effects of V-125 and bexarotene in HER2^+^ breast cancer models. **Methods:** Female MMTV-Neu mice bearing mammary tumors were treated with control, V-125 (100 mg/kg diet), or bexarotene (100 mg/kg diet) for 10 days. RNA sequencing was used to identify differentially expressed genes and pathways. Candidate targets were validated by qPCR and immunohistochemistry (IHC). Immune modulation was evaluated by IHC staining for CD8 cells and CD206^+^ macrophages in tumors to capture the tumor microenvironment. Functional assays in JIMT-1 human HER2^+^ cells assessed RXR target activation and clonogenic potential in tumor cells. **Results:** V-125 induced broader transcriptional changes than bexarotene, including selective upregulation of *Nrg1*, *Nfasc*, *Lrrc26*, and *Chi3l1* genes associated with improved patient survival. Pathway analysis revealed regulation of immune activation, cancer signaling, and lipid metabolism. Both V-125 and bexarotene suppressed colony formation in JIMT-1 cells, confirming previous observations about RXR-dependent inhibition of tumor cell growth. Moreover, V-125 in vivo had distinct capabilities to increase CD8 cell infiltration and reduced CD206^+^ macrophages, whereas bexarotene did not. **Conclusions:** V-125 but not bexarotene reprograms tumor transcriptional programs and the immune landscape in an anti-tumor manner in the MMTV-neu mouse model and in in vitro models of HER2^+^ breast cancer. This highlights its promise as a selective RXR agonist with anti-tumor and immunomodulatory activity in HER2^+^ breast cancer.

## 1. Introduction

Modulating cellular function through transcriptional control is a promising therapeutic strategy for a wide range of diseases, including cancer. Central to this process are nuclear receptors, a family of ligand-activated transcription factors that regulate gene expression programs governing cell growth, metabolism, and differentiation. Among them, the retinoid X receptor (RXR) holds particular significance due to its ability to form heterodimers with nearly one-third of all known human nuclear receptors [1,2].

RXRs exist as three subtypes (α, β, and γ), each with distinct but overlapping patterns of tissue expression [3,4,5]. Their broad distribution across tissues underscores their integral role in regulating physiological processes including lipid metabolism, inflammation, and cell fate determination [5,6]. Consequently, RXRs have emerged as attractive therapeutic targets in cancer, metabolic, inflammatory, and neurodegenerative disorders [7,8].

In breast cancer, elevated RXRα expression has been associated with improved overall survival, highlighting its potential as a therapeutic target [9]. Upon ligand binding, such as endogenous vitamin A derivative 9-cis-retinoic acid, RXRs undergo conformational changes that alter cofactor recruitment and DNA binding, leading to the activation or repression of target genes [10]. Through heterodimerization with nuclear receptors such as peroxisome proliferator-activated receptor (PPAR), liver X receptor (LXR), and retinoic acid receptor (RAR), RXR can orchestrate transcriptional programs that suppress tumor growth by inhibiting cell proliferation, promoting apoptosis and differentiation, altering metabolism, or modulating immune response [5,11,12,13,14,15]. While several synthetic RXR agonists have been developed [16,17,18], bexarotene (Targretin) remains the only FDA-approved RXR agonist, currently in use to treat cutaneous T-cell lymphoma [19]. Bexarotene has been investigated in clinical trials for lung and breast cancers [20,21,22]. However, its broader clinical use has been limited due to significant side effects, including hypertriglyceridemia and hypothyroidism, largely attributed to its cross-reactivity with RARs [21].

To overcome the limitations of bexarotene, numerous synthetic RXR ligands have been developed. Several of these compounds have shown considerable promise in preclinical cancer models for both treatment and prevention [16,17,23,24,25]. While RXR serves as the direct target of these compounds, their downstream effects on gene expression can vary based on cellular context, RXR subtype selectivity, and tumor type. Furthermore, the concentration of ligand also influences RXR activity by altering dimer preferences, leading to distinct transcriptional programs across tissues [26,27,28]. For instance, one of our recent studies compared the effects of MSU-42011, a novel RXRα agonist, and bexarotene on the transcriptome in mammary tumors of HER2^+^ mouse mammary tumor virus (MMTV)-Neu mice [24]. Although both compounds modulated overlapping pathways, they exhibited distinct transcriptional effects in areas such as focal adhesion, extracellular matrix remodeling, and immune signaling, highlighting the complexity of RXR biology, which suggests that exploring the differential effects of RXR agonists on gene transcription may provide valuable insights into their therapeutic potential for cancer treatment.

Beyond direct effects on cancer cells, RXR agonists show great promise in shaping the tumor microenvironment by increasing anti-tumor immune populations and reducing pro-tumor immune cells [15,16,29,30,31,32]. Among these immune cells, macrophages and T cells are especially critical in breast cancer progression and immune modulation [33,34,35,36]. RXR plays a central, context-dependent role in macrophage biology, influenced by factors like tissue type and whether macrophages are resident or infiltrating [15,31,37]. Deletion of RXRα in hematopoietic progenitor cells significantly impairs macrophage populations in the liver, lung, spleen, and peritoneal cavity, highlighting RXR’s vital role in maintaining tissue-specific macrophage [37]. Prior work has demonstrated that RXR activation can reprogram tumor-associated macrophages (TAMs) to a tumor-suppressing phenotype [15,29,31,38]. Emerging evidence suggests that RXRs also influence T cell biology. RXR signaling has been shown to regulate T cell activation, differentiation, and trafficking [13]. RXR heterodimerization with partners such as PPARs or LXRs can impact T cell metabolic programming and effector function, suggesting that RXR agonists may modulate anti-tumor T cell responses [39,40].

To address the limitations of earlier RXR agonists, V-125, a next-generation RXR agonist was developed with improved pharmacokinetic properties, greater selectivity, and reduced toxicity [18]. In murine models of breast and lung cancer, V-125 effectively inhibits tumor growth [38]. Furthermore, V-125 appears more selective than other RXR agonists, potentially reducing off-target effects [18,38]. Moreover, V-125 downregulates genes involved in proliferation and survival while upregulating those related to differentiation and apoptosis, making it a promising candidate for targeting RXR signaling in cancer [38]. In this study, RNA sequencing was performed to compare the molecular pathways activated by the novel compound V-125 and FDA-approved drug bexarotene. By analyzing the transcriptomic profiles, we aimed to identify distinct and overlapping pathways influenced by each treatment. Selected genes showing significant differential expression were further validated using quantitative PCR (qPCR) and immunohistochemistry (IHC) to confirm their expression at both the mRNA and protein levels. To assess the compounds’ impact on the immune landscape, IHC was also used to evaluate markers of tumor-promoting CD206^+^ macrophages and tumor-suppressing CD8 cells, enabling quantification of immune cell subsets within the tumor microenvironment. This integrated transcriptomic and functional analysis allowed us to gain a comprehensive understanding of the gene regulatory and immunomodulatory effect of V-125 in comparison to bexarotene, providing valuable insights into the therapeutic potential and the molecular mechanisms underlying the effects of these compounds in the context of breast cancer.

## 2. Materials and Methods

### 2.1. Cell Culture

The human HER2^+^ breast cancer cell line JIMT-1 (AddexBio, San Diego, CA, USA) was originally established from the pleural metastasis of a patient with trastuzumab-resistant breast cancer [41]. The cells were cultured in Dulbecco’s Modified Eagle’s Medium (DMEM) (Corning Inc., Corning, NY, USA), supplemented with 10% heat-inactivated Fetal Bovine Serum (FBS) (Gibco, Thermo Fisher Scientific, Waltham, MA, USA), 1% Penicillin-Streptomycin solution (Corning Inc., Corning, NY, USA) and 1% L-glutamate (Gibco, Thermo Fisher Scientific, Waltham, MA, USA) to support optimal cell growth and viability. To culture E18-14C-27 cell line we also used the DMEM media, supplemented with 10% heat-inactivated FBS and 1% penicillin-streptomycin. Both JIMT-1 and E18-14C-27 cells were maintained in a humidified incubator at 37 °C with 5% CO_2_.

### 2.2. Drugs

The synthesis and preparation of V-125 were previously described [38,42]. Bexarotene was purchased from LC Laboratories (Woburn, MA, USA). For in vivo studies, both V-125 and bexarotene were separately dissolved in a vehicle consisting of one part ethanol and three parts highly refined coconut oil (Neobee oil, Thermo Fisher Scientific, Waltham, MA, USA). Using a stand mixer (KitchenAid, Benton Harbor, MI, USA), 1 kg of powdered 5002 rat chow (PMI Nutrition, St. Louis, MO, USA) was mixed with 50 mL vehicle or drug dissolved in vehicle. A 100 mg/kg diet (~25 mg per kg per day body weight) was given to the MMTV-Neu mice. The mice always had free access to the diet and were allowed to feed ad libitum.

### 2.3. In Vivo Experiments

MMTV-Neu mice [43] from our breeding colony (founders bought from Jackson Laboratory, Bar Harbor, ME, USA) were fed pelleted chow. Mice were palpated twice a week for the presence of new tumors. Once tumors were identified, mice 5002 powder chow feeding was initiated. Tumors were measured twice weekly with a caliper until they reached ~5 mm in diameter, at which point mice were randomly assigned to a control diet or an RXR agonist diet of 100 mg per kg per day for 10 days. Tumor volume was then measured on days 0, 3, 7, and 10 following the initiation of treatment to assess treatment-dependent growth responses. On day 10, tumors were harvested, weighted and sectioned to either flash frozen for RNA-seq and qPCR, or preserved for IHC in neutral buffered formalin.

### 2.4. RNA Sequencing

Frozen tumors (*n* = 4) from each treatment group were weighed (30 mg) and homogenized. Total RNA was extracted using the RNeasy Mini Kit (Qiagen, Hilden, Germany), and RNA quality was assessed by Novogene (Sacramento, CA, USA). As previously described [44], RNA sequencing was performed by Novogene (Sacramento, CA, USA). Raw sequencing reads were processed to obtain read counts, and differential expression analysis was conducted using the limma package in R (RStudio 2024.04.0 with R v4.3.2.) [45]. Enrichment analysis was performed using EnrichR [46] and Ingenuity Pathway Analysis (IPA) (Qiagen) [47]. Genes with cancer-associated functions were identified from IPA results, and those with a log fold change (logFC) < −1 or > 1 and an adjusted *p*-value (adj. *p*) < 0.05 were retained as initial candidates. To refine the candidate gene list, we applied additional selection criteria, and genes were evaluated for their association with patient overall survival using Kaplan–Meier (KM) Plotter [48] and their potential involvement in macrophage biology through TIMER2.0 analysis [49]. Reference datasets used for pathway enrichment included Gene Ontology (GO) [50], WikiPath [51], and Reactome [52] databases. Additionally, potential roles in protein–protein interaction networks were explored through STRING analysis [53]. Finally, all shortlisted genes were cross-referenced with supporting evidence from peer-reviewed literature to ensure both mechanistic plausibility and clinical significance. This multi-tiered approach ensured that shortlisted genes were not only differentially expressed but also biologically relevant, clinically significant, and mechanistically linked to cancer pathogenesis.

### 2.5. Real-Time Quantitative PCR (RT-qPCR)

Total RNA was extracted from frozen tumor sections or treated JIMT-1 cells using the RNeasy Mini Kit (Qiagen) according to the manufacturer’s protocol. RNA concentration and purity were assessed using a NanoDrop spectrophotometer (Thermo Fisher Scientific, Waltham, MA, USA), and samples were normalized to equal concentrations prior to cDNA synthesis. For JIMT-1 cell experiments, 1 × 10^6^ cells were seeded in 6-well plates and allowed to adhere for 24 h before treatment with V-125 or bexarotene (600 nM). After an additional 24 h of incubation, cells were harvested for RNA extraction. 500 ng of RNA was used for cDNA synthesis using a High-Capacity cDNA Reverse Transcription Kit (Applied Biosystems, Foster City, CA, USA). Quantitative PCR was performed on a QuantStudio 7 Flex Real-Time PCR System (Thermo Fisher Scientific, Waltham, MA, USA) using SYBR Green Master Mix. The delta-delta CT method was used to analyze relative quantification of the target gene expression, with mouse Glyceraldehyde-3-phosphate dehydrogenase (GAPDH) serving as an internal control for tumor RNA and human GAPDH used for JIMT-1 cell samples. Data are presented as fold change relative to control, and error bars represent the standard error of the mean (SEM) from biological replicates. Primer sequences (Integrated DNA Technologies, Coralville, IA, USA) are listed in Table 1.

### 2.6. Immunohistochemistry

Formalin-fixed tissues were embedded in paraffin and sectioned by the Indiana University, Indianapolis Histology Core. Paraffin-embedded tumor sample sections were deparaffinized into xylene three times for 5 min each, then rehydrated with a graded series of ethanol. Antigen retrieval was achieved by boiling slides in citrate buffer for 10 min. After washing the slides in ddH_2_O, endogenous peroxidase activity was blocked by incubating sections in 3% hydrogen peroxide for 10 min at room temperature. The samples were permeabilized either in PBS or 1× Tris-buffered saline with 0.1% Tween for 15 min, and nonspecific binding was inhibited by 10% normal goat or animal free serum for 60 min at room temperature. Tissue sections were stained with primary antibodies against Nrg1 (1:400 dilution, PA5-120049, Thermo Fisher Scientific), Nfasc (1:500 dilution, 26351-1-AP, Proteintech, Rosemont, IL, USA), Slit2 (1:400 dilution, PA5-31133, Thermo Fisher Scientific), Chi3l1 (1:200 dilution, 12036-1-AP, Thermo Fisher Scientific), CD206 (1:500 dilution, 18704-1-AP, Thermo Fisher Scientific) and CD8 (1:100 dilution, 70306, Cell Signaling Technology, Danvers, MA, USA). Primary antibodies were left on tissues overnight at 4 °C in a humidifying chamber. Following three PBS washes, the sections were tagged with biotinylated secondary antibodies (anti-rabbit, Cell Signaling Technology) for 30 min at room temperature. Again, after three washes with PBS, the 3, 5-diaminobenzidine (DAB) substrate (Cell Signaling Technology) was employed to detect signals, and sections were counterstained with hematoxylin (Vector Laboratories, Newark, NJ, USA). Quantification of DAB-positive staining was performed using the Fiji image processing program (version 2.7.0) (ImageJ2). For each tumor, regions of interest (ROIs) were manually selected within tumor tissue at 40× magnification. ROIs were chosen consistently across samples by selecting comparable tumor areas while avoiding tissue folds, or artifacts. DAB signal was quantified using the Color Deconvolution plugin, which separates the brown DAB channel from hematoxylin counterstaining. After deconvolution, the DAB channel image was converted to an 8-bit grayscale image and subjected to a uniform threshold to isolate positive staining. The mean gray value within each ROI was then measured and converted to optical density (OD) using the standard formula OD = log (max intensity/mean intensity, with a maximum intensity of 255 for 8-bit images) [54,55]. Multiple ROIs were analyzed per tumor, and results were averaged to generate one value per biological replicate. Data is presented as mean ± SEM for each treatment group.

### 2.7. Kaplan–Meier Plot Generation

Kaplan–Meier Plotter was used to perform survival analysis (https://kmplot.com/analysis/, accessed on 8 October 2025). The KM plotter uses gene expression and survival data from databases such as GEO, EGA, and TCGA. By analyzing publicly available patient data, it can determine the effect of any gene breast cancer survival, and various other tumor types [48]. KM plotter divides patient data based on the expression levels of the specified gene (high versus low). The findings are presented as a survival plot, which depicts the probability of survival over time for each patient group (high versus low gene expression). Furthermore, hazard ratio and log rank *p*-values are provided to indicate statistical significance. In our study, breast cancer data was used to generate KMPlot of overall survival for the target genes.

### 2.8. Colony Formation Assay

JIMT-1 cells were seeded at a density of 2000 cells per well in 6-well tissue culture plates (Corning) and allowed to adhere overnight. After 24 h, cells were treated with vehicle control, V-125 (600 nM), or bexarotene (600 nM). Drug treatments were refreshed twice weekly to maintain consistent exposure for 14 days. Colony formation was assessed in E18-14C-27 cells treated with vehicle control, V-125 (600 nM), or bexarotene (600 nM). Cells were seeded at 500 cells per well in 6-well plates and allowed to grow for 14 days. After 14 days, culture medium was aspirated, and cells were gently rinsed with phosphate-buffered saline (PBS). Colonies were fixed with 4% paraformaldehyde (PFA) for 15 min at room temperature, washed twice with PBS, and stained with 0.5% (*w*/*v*) crystal violet (prepared in 25% methanol) for 30 min. Each well was rinsed with water to remove excess stain, and plates were air-dried overnight. Colony formation was quantified using ImageJ software, where stained colonies were counted and analyzed for area and intensity.

### 2.9. Statistical Analysis

Statistical analyses were performed using GraphPad Prism Version 8.0.2 (San Diego, CA, USA). One-way ANOVA was used to compare more than two groups and significant differences between groups were determined by the Tukey HSD multiple comparisons test. For analyses involving tumor-volume measurements collected across multiple time points and treatment groups, a two-way repeated-measures ANOVA was performed with treatment and time as factors. When a significant main effect or interaction was detected, Tukey’s post hoc multiple comparisons test was applied. This analysis accounts for the matched nature of longitudinal measurements collected from the same mice over time. Statistical significance was defined as *p* < 0.05 for all tests. Error bars represent the standard error of the mean (SEM) for the number of biological replicates indicated by “*n*” in each figure. Figure legends specify the statistical tests used and sample sizes for each experiment. Differentially expressed genes from RNA-sequencing analysis were identified using the toptable function in limma, applying Benjamini–Hochberg false discovery rate (FDR) correction.

## 3. Results

### 3.1. V-125 and Bexarotene Regulate Overlapping and Distinct Cancer-Associated Gene Sets

To define the transcriptional and functional effects of the RXR agonist V-125 in HER2^+^ breast cancer, we employed a multi-step workflow (Figure 1A). MMTV-Neu mice (four per group) bearing mammary tumors ~5 × 5 mm in diameter were randomized to receive either control diet (i.e., placebo), V-125 (100 mg/kg diet), or bexarotene (100 mg/kg diet) for 10 days. Tumors were then harvested and sent for RNA-sequencing. The RNA-sequence analysis revealed that V-125 induced broader transcriptional changes than bexarotene, with 71 uniquely regulated genes compared to 17 for bexarotene, and 15 shared targets (Figure 1B). This indicates that although both agonists act through RXR, V-125 elicits a more extensive and distinct regulation of genes.

From the overlapping and unique gene sets, we curated cancer-associated candidates using IPA functional annotations, survival data, and literature evidence (Figure 1C). These gene candidates included *Nrg1* (apoptosis signaling) (adj. *p*: 0.0047, logFC: 3.1088 for V-125; adj. *p*: 0.0430, logFC: 2.3207 for bexarotene), *Nfasc* (cell adhesion) (adj. *p*: 0.0065, logFC: 2.1538), *Chi3l1* (tumor growth) (adj. *p*: 0.0208, logFC: 1.1429), and *Lrrc26* (invasion/metastasis) (adj. *p*: 0.0274, logFC: 1.0882). Among the selected gene list, *Nrg1* is the only gene commonly up-regulated by both V-125 and bexarotene. Notably, while *Nrg1* was the only gene in the RNAseq dataset that was commonly up-regulated by both RXR agonists, its expression was strongly induced by V-125. Further analysis using volcano plots highlighted the magnitude of the gene expression changes. The volcano plot for V-125 vs. control showed a clear cluster of highly significant genes with large fold changes (Figure 1D). In contrast, the bexarotene vs. control volcano plot displayed a less pronounced effect, with fewer genes showing significant changes in expression (Figure 1E).

### 3.2. V-125 Upregulates Genes Associated with Improved Patient Survival

As shown in Figure 2A, higher expression of *LRRC26* is associated with improved overall survival in breast cancer patients. Consistent with this observation, V-125 significantly upregulated *Lrrc26* expression (Figure 2B). Bexarotene also elevated *Lrrc26* levels but to a substantially lower degree. KMPlot analysis further demonstrated that elevated *NRG1* expression correlates with favorable survival in HER2^+^ breast cancer patients (Figure 2C). Consistent with this clinical association, V-125 markedly upregulated *Nrg1*, whereas bexarotene did not produce a detectable increase (Figure 2D). Similarly, V-125 markedly increased the expression of *Nfasc* and *Chi3l1* (Figure 2F,H), while bexarotene induced substantially weaker expression. Importantly, higher expression of these genes, particularly *NFASC* and *CHI3L1*, is significantly associated with prolonged HER2^+^ breast cancer patient survival, as shown in the KMPlot analyses (Figure 2E,G). Together, these findings indicate that V-125 not only induces greater magnitude of gene expression changes than bexarotene but also preferentially activates genes that are clinically linked to improved patient outcomes.

### 3.3. RXR Agonists Modulate Cancer-Relevant Pathways in MMTV-Neu Mammary Tumors

To gain mechanistic insight into the biological processes regulated by RXR agonists, we performed pathway enrichment analysis using Enrichr on all the differentially expressed genes (DEGs), both upregulated and downregulated from MMTV-Neu mammary tumors treated with either V-125 or bexarotene, compared to control. The DEGs included both upregulated and downregulated genes and were analyzed using the Gene Ontology (GO) Biological Process 2025, WikiPathways Mouse 2024, and Reactome Pathways 2024 databases (Figure 3).

Pathway enrichment analysis revealed significant modulation of pathways associated with lipid metabolism and immune responses. The most significantly enriched pathway was the inflammatory response (GO:0006954), indicating an immunomodulatory effect of treatment. Consistent with the established role of RXR agonists in metabolic regulation, multiple lipid and steroid-related pathways were also prominently altered, including regulation of cholesterol biosynthesis by SREBP (Reactome), cholesterol biosynthesis (WP103), metabolism of steroids (Reactome), activation of gene expression by SREBF (Reactome), and fatty acid biosynthesis (WP336). These findings highlight the contribution of RXR agonists to lipid metabolic reprogramming in tumors. Several of these pathways are directly linked to RXR’s role in regulating lipid homeostasis through heterodimerization with LXR and PPAR. In support of this, the PPAR signaling pathway (WP2316) was significantly enriched, consistent with RXR-PPAR-mediated transcriptional regulation. Additional enriched pathways included omega-9 fatty acid synthesis (WP4351), and retinol metabolism (WP1259), further illustrate the broad metabolic influence of RXR activation. Additionally, V-125 and bexarotene modulated immune-associated processes such as neutrophil degranulation (Reactome) and activation of NF-κB-inducing kinase activity (GO:0007250), implicating RXR signaling in immune regulation within the tumor microenvironment. Pathways related to regulation of extrinsic apoptotic signaling (GO:2001240, GO:2001239) and cellular response to hypoxia (GO:0071456) were also differentially regulated, suggesting broader effects on tumor cell survival. Furthermore, processes involved in blood coagulation (GO:0030193) and regulation of blood vessel endothelial cell proliferation involved in sprouting angiogenesis (GO:1903589) were enriched, indicating a broader impact on tumor homeostasis and vascular support system. Collectively, these findings demonstrate that V-125 and bexarotene treatment alters a diverse set of pathways and reshapes tumor transcriptional programs through coordinated regulation of immune signaling, lipid metabolism and cancer-related signaling.

### 3.4. Overlapping Function of V-125 and Bexarotene

Prior evidence indicates that the RXR agonist IRX4204 inhibits the growth of HER2^+^ breast cancer cells and can be applied to HER2 inhibitor–resistant JIMT-1 human breast cancer cells [56]. Given these findings, we next evaluated whether V-125 and bexarotene exhibit similar functional activity in HER2^+^ tumor models. We performed functional assays to determine whether the application of V-125 and bexarotene also translated into measurable effects on JIMT-1 tumor cell viability through measurement of clonogenic potential. To validate RXR target engagement, we first examined the expression of canonical RXR-responsive genes *Abca1* and *Srebp* in JIMT-1 cells treated with V-125 or bexarotene. Both RXR agonists significantly upregulated *Abca1* and *Srebp* mRNA levels compared to vehicle-treated controls (Figure 4A,B), confirming effective activation of RXR-dependent transcriptional programs. To evaluate the functional consequences of these transcriptional changes, we performed a colony formation assay using JIMT-1 cells. Consistent with prior findings seen with IRX4204, both V-125 and bexarotene significantly reduced the colony size and number compared to vehicle-treated controls, indicating a robust suppression of long-term clonogenic potential (Figure 4C,D). To further explore whether this effect extended to the MMTV-Neu in vivo system, we examined tumor growth over time. Dietary delivery of V-125 produced a pronounced reduction in tumor volume over 10 days compared to control-fed mice, whereas bexarotene showed no inhibitory effect (Figure 4E). We also tested the murine E18-14C-27 cell line derived from the MMTV-Neu system. Contrary to the results seen in JIMT-1 cells, neither V-125 nor bexarotene had an effect on colony formation (Appendix A). These results suggest that the functional response to RXR agonists may be cell line-specific, likely reflecting intrinsic biological differences between HER2^+^ models. Notably, JIMT-1 cells are known to be stem-cell like and trastuzumab-resistant, features that may contribute to their heightened sensitivity to RXR-targeting compounds compared with the more differentiated MMTV-Neu derived E18-14C-27 cells. Together, these findings confirm that V-125 and bexarotene effectively activate RXR signaling and can play context-dependent overlapping and distinct roles in HER2^+^ breast cancer.

### 3.5. V-125 and Bexarotene Enrich for Distinct Pathways in MMTV-Neu Mammary Tumors

A direct comparison between V-125 and bexarotene highlighted that V-125 treatment resulted in the upregulation of GO Biological Process pathways including inflammatory response, cellular response to hypoxia, and triglyceride metabolism (Figure 5A, in pink, and Appendix A). Among the most significantly enriched gene sets was the inflammatory response pathway (GO:0006954), suggesting that V-125 may induce an immunostimulatory tumor microenvironment. Supporting this, activation of NF-κB-inducing kinase activity (GO:0007250) and cellular response to hypoxia (GO:0071456) were also significantly upregulated, suggesting the capacity of V-125 to remodel the tumor microenvironment. Importantly, the pathway positive regulation of triglyceride biosynthetic process (GO:0010867) was also significantly enriched in V-125-treated tumors.

In contrast, the top downregulated pathways (Figure 5A, in blue, and Appendix A) in tumors treated with V-125 were predominantly associated with lipid and cholesterol homeostasis, ER stress response, and neural cell development. The most significantly downregulated pathway was negative regulation of protein exit from the endoplasmic reticulum (ER) (GO:0070862), suggesting that V-125 may alleviate ER stress or modify secretory pathway activity. This could influence protein folding and trafficking within the tumor environment. Notably, cholesterol homeostasis (GO:0042632) and sterol homeostasis (GO:0055092) were significantly suppressed, aligning with V-125’s observed ability to downregulate tumor lipid metabolism. In addition, positive regulation of lipid biosynthetic process (GO:0046889) and positive regulation of triglyceride biosynthetic process (GO:0010867) were also significantly downregulated. These findings further reinforce the concept that V-125 inhibits lipogenic reprogramming, a hallmark of aggressive tumor phenotypes [57]. V-125 also suppressed activation of NF-κB-inducing kinase activity (GO:0007250), which appears contradictory given its upregulation in the upregulated pathway analysis. This likely reflects temporal or cell-type-specific regulation where V-125 may simultaneously prime immune components for activation while dampening NF-κB signaling in tumor or stromal compartments to prevent chronic inflammation. Furthermore, pathways associated with neural crest cell development (GO:0014032) and cellular response to peptide hormone stimulus (GO:0071375) were also significantly downregulated, which may represent broader differentiation or signaling programs being repressed in the tumor microenvironment.

Bexarotene treatment on the other hand predominantly upregulated pathways involved in cellular signaling, differentiation, and metabolic regulation, with limited activation of immune processes (Figure 5B, in pink, and Appendix A). The most significantly enriched pathway was negative regulation of transport (GO:0051051), suggesting a role in modulating intracellular trafficking which may alter vesicle or protein movement, potentially impacting cellular signaling and metabolism in the tumor microenvironment. Bexarotene also upregulated ERBB4 signaling (GO:0038130) and activation of transmembrane receptor protein tyrosine kinase activity (GO:0007171), pointing to enhanced receptor-mediated communication. Interestingly, pathways associated with chronic inflammatory response (GO:0002544) and inflammatory response to wounding (GO:0090594) were also upregulated. This pattern may represent a more reparative or tissue-remodeling inflammatory state rather than acute immune activation.

In contrast, bexarotene downregulated a different set of pathways, prominently involving cytokine production and steroid hormone responses (Figure 5B, in blue, and Appendix A). Positive regulation of tumor necrosis factor superfamily cytokine production (GO:1903557) was significantly downregulated, supporting bexarotene’s role in dampening pro-inflammatory immune responses. Bexarotene also suppressed negative regulation of extrinsic apoptotic signaling pathways in the absence of ligand (GO:2001240) and negative regulation of signal transduction in absence of ligand (GO:1901099). Additional downregulated pathways include regulation of protein processing (GO:0070613), response to steroid hormone (GO:0048545), negative regulation of small molecule metabolic process (GO:0062014) and regulation of transmembrane receptor protein serine/threonine kinase signaling (GO:0090101), suggesting broader metabolic reprogramming and altered receptor-mediated/endocrine signaling pathways.

### 3.6. V-125 Promotes an Anti-Tumor Immune Microenvironment

To determine whether the differences in transcriptional changes were translated into functional immune modulation, particularly the observation that the pathway analysis predicts that V-125 induces an immunostimulatory state, we examined the immune cell composition within tumors by IHC. Quantitative analysis revealed a significant increase in CD8 cell presence in V-125-treated tumors compared to both vehicle control and bexarotene-treated groups (Figure 6A,B). This suggests that V-125 promotes an immunostimulatory tumor microenvironment that favors CD8 cell presence. In contrast, analysis of CD206^+^ staining, which is a marker of M2-type tumor associated macrophages, demonstrated a significant decrease in V-125-treated tumors, whereas bexarotene treatment had no such effect (Figure 6C). This analysis is supported by representative IHC images showing reduced CD206 staining in the V-125 group and increased staining in bexarotene-treated tumors (Figure 6D). The inverse relationship between CD8 cell presence and CD206^+^ macrophage abundance suggests a functional reprogramming of the tumor immune microenvironment by V-125. CD206 is a well-established marker of tumor-promoting macrophages which suppress CD8 cell activity and promote tumor progression [58,59]. Thus, V-125 may enhance anti-tumor immunity not only by recruiting CD8 T cells but also by depleting tumor-promoting CD206^+^ macrophages. Conversely, the abundance of CD206^+^ macrophages in bexarotene-treated tumors may contribute to reduced CD8 T cell presence.

To explore potential molecular mediators of this immune response, we interrogated our RNA-seq data and prioritized *Slit2*, a secreted axon guidance molecule with emerging roles in immune regulation [60,61]. We performed qPCR and confirmed a robust induction of *Slit2* expression in V-125-treated tumors compared to control and bexarotene (Figure 6E). Moreover, survival analysis using patient data revealed that high *SLIT2* expression was associated with significantly improved overall survival in breast cancer (Figure 6F), highlighting its potential clinical relevance and consistent with a differential effect of V-125 on regulation of immune cells in the tumor microenvironment.

## 4. Discussion

RXR has long been recognized as a compelling therapeutic target in oncology because of its ability to function as a master regulator of gene expression by forming homodimers or heterodimers with a wide array of other nuclear receptors [16,62,63]. While RXR agonists share the same target and display anti-tumor activity, their downstream gene expression profiles and clinical effects vary widely [24]. These diversities likely arise from compound-specific interactions with RXR that influence dimer partner selection, co-activator recruitment, and chromatin accessibility, ultimately leading to distinct biological outcomes [3,64]. Our study sought to understand these differences by comparing the effects of the next-generation RXR agonist V-125 with the FDA-approved agonist bexarotene in MMTV-Neu HER2^+^ mouse breast cancer model.

Using RNA sequencing, we directly compared the pathway activation and biological activity of V-125 and bexarotene. We found that V-125 exerts a more therapeutically relevant transcriptional profile than the clinically approved RXR agonist bexarotene. V-125 modulated a larger set of cancer-associated genes, including, *Lrrc26*, *Nrg1*, *Nfasc*, and *Chi3l*. Among these, *Nrg1* encodes a ligand for ERBB receptors and has been implicated in apoptosis and immune modulation [65,66,67,68], while *Chi3l1* (YKL-40) regulates tissue remodeling and macrophage recruitment [69,70,71]. Elevated expression of these genes correlated with improved overall survival in breast cancer patient datasets, highlighting their potential clinical relevance. In contrast, bexarotene elicited only modest induction of these targets, suggesting that V-125 engages RXR in a unique manner that preferentially activates tumor-suppressive programs.

Pathway enrichment analysis further reinforced these mechanistic distinctions. V-125 strongly upregulated immune-related processes, including inflammatory response, activation of NF-κB-inducing kinase (NIK) activity, and regulation of extrinsic apoptotic signaling pathways, while simultaneously suppressing cholesterol and sterol biosynthesis. Notably, activation of the NF-κB pathway has been linked to enhanced antitumor immunity, as NIK is required for maintaining the metabolic fitness and effector function of CD8 cells within the tumor microenvironment. Consistent with this, V-125 treatment increased CD8 cell infiltration in MMTV-Neu tumors as observed by IHC, suggesting that the enrichment of NIK activity may contribute to a more immunostimulatory tumor milieu [72]. Additionally, lipid metabolism plays a key role in supporting membrane synthesis and signaling in rapidly proliferating tumor cells [73,74]. The simultaneous activation of immune pathways and repression of lipogenic programs suggests that V-125 can remodel the tumor microenvironment by enhancing anti-tumor immunity and suppressing lipogenic reprogramming, a metabolic hallmark of aggressive tumors [75]. Mechanistically, these effects are consistent with the dual regulatory functions of RXR-PPAR and RXR-LXR heterodimers. RXR-PPAR complexes are central regulators of fatty acid oxidation and lipid catabolism, reducing lipid accumulation and metabolic stress within tumor cells while also promoting anti-inflammatory macrophage polarization and T-cell activation [39,76,77]. Conversely, RXR-LXR heterodimers act as cholesterol sensors that suppress sterol biosynthesis and promote cholesterol efflux, while simultaneously enhancing transcription of genes involved in innate immune activation and antigen presentation [78,79,80,81]. Thus, the transcriptional signature observed with V-125 treatment marked by downregulation of lipid biosynthetic pathways and upregulation of immune signaling likely reflects the coordinated engagement of RXR-PPAR and/or RXR-LXR signaling axes. On the other hand, the suppression of cytokine production by bexarotene suggests that it may foster a more immunosuppressive or tissue-remodeling state compared with the immunostimulatory profile of V-125.

Both V-125 and bexarotene reduced the clonogenic growth of human HER2^+^ breast cancer cells, confirming that RXR activation can impair long-term tumor cell survival. This observation is consistent with previous observations with the RXR agonist IRX4204 [56]. Interestingly, the in vivo tumor-volume analysis in the MMTV-Neu model illustrated distinct effects for V-125. Over a 10-day treatment period, V-125 significantly decreased tumor volume compared to both control and bexarotene, whereas bexarotene did not produce meaningful tumor shrinkage. This divergence in in vivo efficacy, despite similar RXR target engagement in vitro, supports the idea that V-125’s anti-tumor activity is heavily driven by its ability to modulate the tumor immune microenvironment, rather than by direct tumor-cell effects alone. These observations further strengthen the conclusion that V-125 uniquely engages the immune compartment.

Based on the above observation, the transcriptional differences in the RNAseq data and pathway analysis appear to translate into distinct functional outcomes. Only V-125 produced immune effects, which were not observed with bexarotene. Immunohistochemical analysis revealed V-125 significantly increased presence of CD8 T cells and reduced CD206^+^ tumor-promoting macrophages. In contrast, bexarotene treatment increased CD206^+^ macrophages and reduced CD8 T cell presence. This is a critical distinction, as CD206^+^ macrophages are a well-established marker of M2-polarized, tumor-associated macrophages, which promote tumor growth, angiogenesis, and suppress CD8 T cell activity [36,58,59,82,83].

The ability of V-125 to reduce these pro-tumorigenic macrophages while enhancing CD8 cell abundance indicates that this new RXR agonist has potential to be used to “reprogram” the tumor microenvironment. These findings are supported by prior work demonstrating RXR agonists’ ability to reprogram tumor-associated macrophages and enhance anti-tumor immunity [16,29]. This finding is particularly notable given that prior research on bexarotene has yielded a more complex and sometimes contradictory picture regarding its immunomodulatory effects. Some studies have suggested bexarotene’s efficacy in certain lymphomas is due to the suppression of specific chemokines, such as CCL22, derived from M2 macrophages [84]. However, in our breast cancer model, we did not observe a reduction in CD206^+^ macrophage presence following bexarotene treatment, suggesting that its effects on macrophage differentiation in solid tumors may be limited or context-dependent, potentially contributing to its reduced efficacy in this setting. To explore the mechanisms underlying the distinct immunomodulatory activity of V-125, we examined transcriptional changes in treated tumors and identified several immune-regulatory genes that were significantly upregulated compared to controls. Among these, *Slit2*, a secreted glycoprotein known for its roles in axon guidance and emerging immunomodulatory functions, showed strong induction by qPCR validation [85]. Previously published data also supports a tumor-suppressive role for *Slit2*, demonstrating its ability to decrease tumor-promoting CD206^+^ M2 macrophage population in breast cancer mouse model [86]. The association between high *SLIT2* expression and improved overall survival in breast cancer patients further validates its clinical relevance. These data support the idea that V-125 not only acts directly on tumor cells but also remodels the tumor microenvironment to favor immune surveillance.

Together, these findings position V-125 as a potent and mechanistically distinct RXR agonist from bexarotene, capable of broad transcriptional reprogramming, enhancement of anti-tumor immunity, and suppression of lipid metabolic pathways in HER2^+^ breast cancer. By selectively activating genes associated with improved patient survival and promoting CD8 cell presence, V-125 may overcome some of the limitations of first-generation RXR agonists such as bexarotene, which are hindered by metabolic side effects and weaker immune activation [6,87,88]. The transcriptional reprogramming and immune activation observed with V-125 vs. bexarotene despite their shared target, underscore the complexity of RXR signaling. While both compounds bind to RXR, their unique chemical structures may lead to different conformations of the RXR receptor. These conformational changes can influence the dimerization of RXRs with their partners (e.g., RAR, VDR, PPAR) and the recruitment of co-activator or co-repressor proteins, resulting in distinct downstream gene expression programs. Our comprehensive gene expression analysis confirms this, showing that V-125 and bexarotene regulate largely non-overlapping sets of genes. This molecular divergence provides a clear rationale for their divergent effects on the tumor microenvironment.

While our findings uncover major transcriptional and functional differences between V-125 and bexarotene, several questions remain open for future study. A more comprehensive characterization of the immune landscape is warranted. Approaches such as flow cytometry and single-cell RNA sequencing could delineate distinct T cell subsets (e.g., CD4^+^, regulatory, and exhausted CD8^+^ populations) as well as macrophage polarization states (M1 vs. M2) to more precisely define how V-125 modulates the tumor immune microenvironment. Moreover, the observed concurrent downregulation of sterol and lipid homeostasis pathways and reduction in CD206^+^ macrophages for V-125 treatment suggests a potential link between lipid metabolism and macrophage activation. Future studies should aim to determine whether metabolic reprogramming can directly influence macrophage polarization or T-cell activation. Future studies will be needed to define the contribution of specific RXR subtypes and heterodimer partners. Also given the immunostimulatory signature of V-125, it will also be important to assess whether this compound synergizes with immune checkpoint blockade or other immunotherapies.

In summary, our study demonstrates that the RXR agonists V-125 and bexarotene, despite sharing a target, have fundamentally different effects, where the next-generation RXR agonist V-125 exerts a broader and more therapeutically relevant transcriptional effect than the FDA-approved RXR agonist bexarotene in a murine HER2^+^ breast cancer model. Using RNA sequencing, pathway enrichment analysis, and functional assays, we show that V-125 not only induces a distinct set of cancer-related genes but also remodels the tumor immune microenvironment to favor anti-tumor immunity. These findings also highlight the therapeutic promise of selective RXR activation in breast cancer and provide mechanistic insight into the downstream pathways regulated by V-125.

## Figures and Tables

**Figure 1 biomedicines-14-00080-f001:**
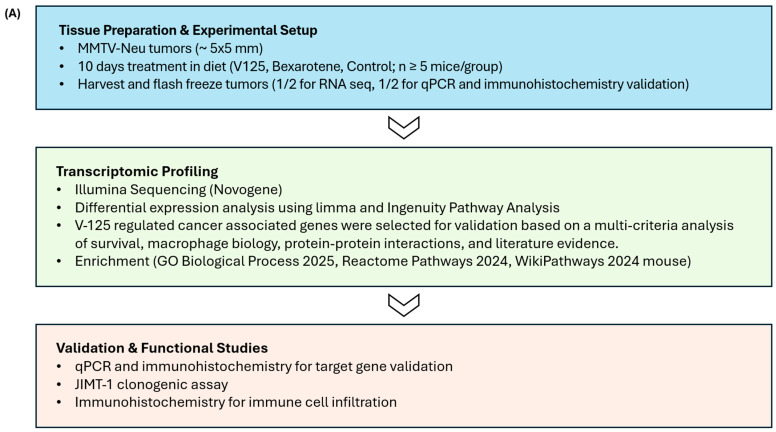
**Comparison of gene expression profiles following V-125 and bexarotene treatment.** (**A**) Experimental flow diagram of the study. (**B**) Venn diagram illustrates the number of differentially expressed genes (DEGs) following treatment with V-125 (V-125 vs. Control) and bexarotene (Bex vs. Control). The overlap shows the number of DEGs common to both treatment groups. (**C**) A table listing selected genes that were differentially expressed, along with their known cancer-related functions. (**D**) Volcano plot showing the DEGs for V-125 treatment compared to the control. The x-axis shows the log fold change (logFC), with positive values indicating upregulation and negative values indicating downregulation. The y-axis shows the negative log10 *p*-value (−log10(adj. *p*-value)), with higher values indicating greater statistical significance. (**E**) Volcano plot of the DEGs for bexarotene treatment compared to the control.

**Figure 2 biomedicines-14-00080-f002:**
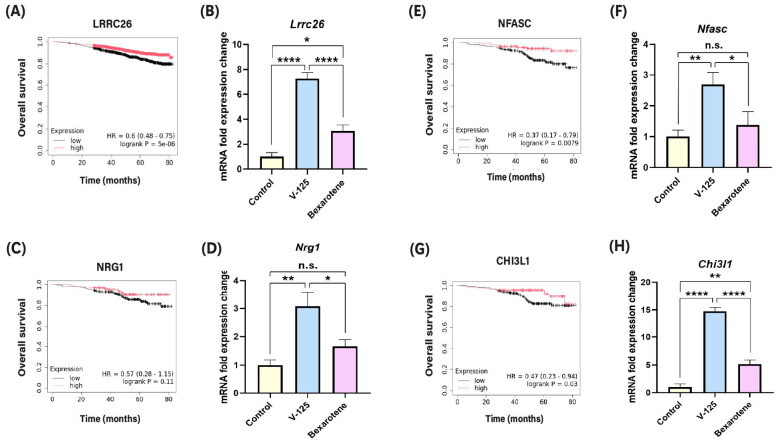
**V-125 regulates genes linked to cancer and immune response in MMTV-Neu mammary tumors.** In MMTV-Neu mammary tumors, selected genes that are differentially expressed in response to V-125 are correlated with breast cancer patient survival. Kaplan–Meier (KMPlot) survival curves correlate expression of the differentially expressed selected genes for V-125 vs. control-treated tumors with overall survival in breast cancer patients [*n* = 2976]; 2100 high 876 low in (**A**) and in HER2^+^ breast cancer patients [*n* = 379]; 135 high 244 low in (**C**), 137 high 242 low in (**E**) and 139 high 240 low in (**G**). Data produced by KMPlot (http://www.kmplot.com, accessed on 8 October 2025). V-125 (100 mg/kg in diet), bexarotene (100 mg/kg in diet) and control treated tumors were harvested and flash-frozen. After homogenizing the tumors, RNA was extracted. (**B**,**D**,**F**,**H**) qPCR was performed using QuantStudio 6 Pro to quantify mRNA expression. * *p* < 0.05, ** *p* < 0.01, **** *p* < 0.0001, n.s. = not significant; *n* ≥ 3 mice/group.

**Figure 3 biomedicines-14-00080-f003:**
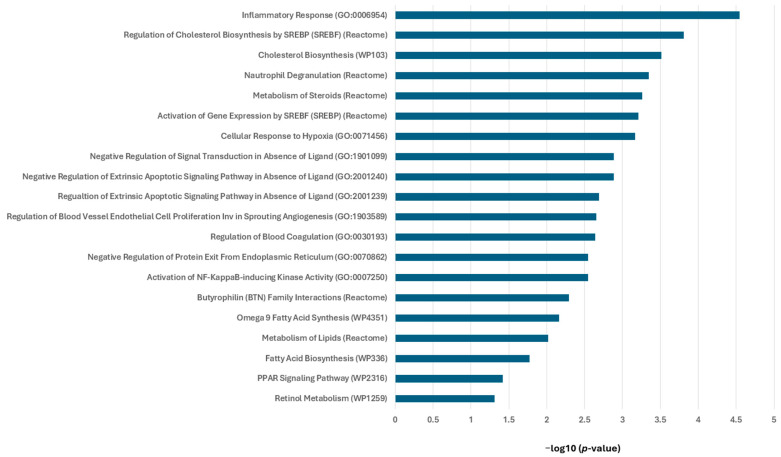
**RXR agonist V-125 and bexarotene regulate cancer-relevant pathways in treated mammary tumors of a HER2^+^ mouse model.** Female MMTV-Neu mice with mammary tumors measuring ~5 mm in diameter were treated either a control diet, V-125 (100 mg/kg diet) or bexarotene (100 mg/kg diet) for 10 days. Differentially expressed genes from V-125 and bexarotene treated vs. control tumors were analyzed using Enrichr. The top enriched pathways are shown as bar graphs for GO Biological Process 2025, WikiPathways Mouse 2024, and Reactome Pathways 2024 databases. Pathway enrichment is plotted as −log10(*p*-value).

**Figure 4 biomedicines-14-00080-f004:**
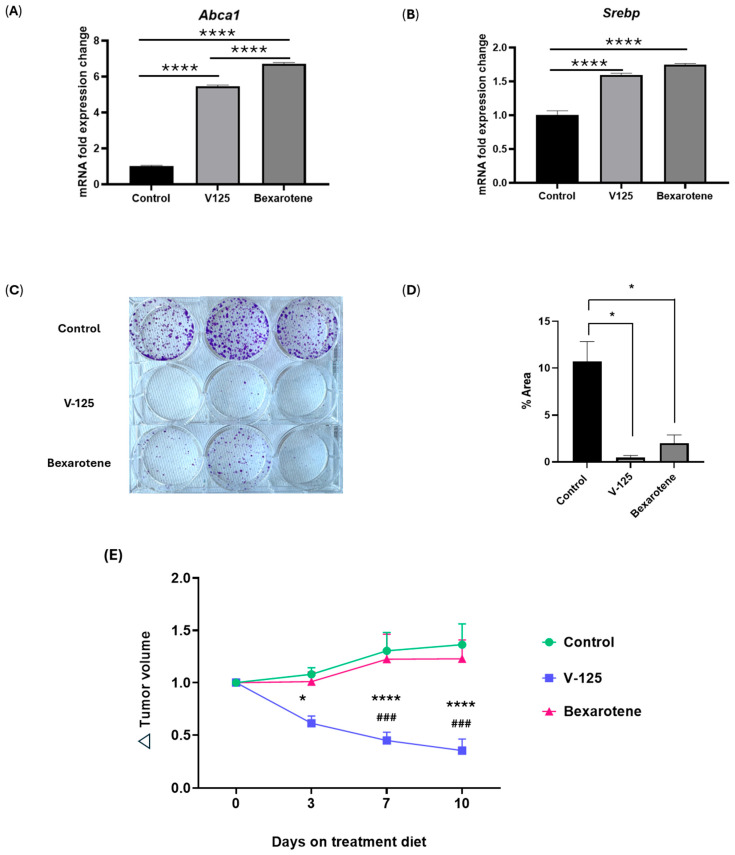
**Both V-125 and bexarotene activate RXR signaling and inhibit colony formation in JIMT-1 HER2**^+^**breast cancer cells.** Quantitative RT-PCR analysis of *Abca1* (**A**) and *Srebp* (**B**) mRNA expression in JIMT-1 breast cancer cells treated with vehicle control, V-125 (600 nM), or bexarotene (600 nM) for 24 h. (**C**) Images of crystal violet stained colonies in 6-well plates after treatment with control, V-125, or bexarotene. (**D**) Quantification of colony area (% area) shows a significant reduction in colony formation in cells treated with V-125 and bexarotene compared to control. (**E**) Tumor volume change (Δ volume) in MMTV-Neu mice fed control diet, V-125 diet, or bexarotene diet over 10 days. Data are shown as mean ± SEM. * *p* < 0.05, **** *p* < 0.0001 versus control, ^###^
*p* < 0.001 versus bexarotene; *n* ≥ 3.

**Figure 5 biomedicines-14-00080-f005:**
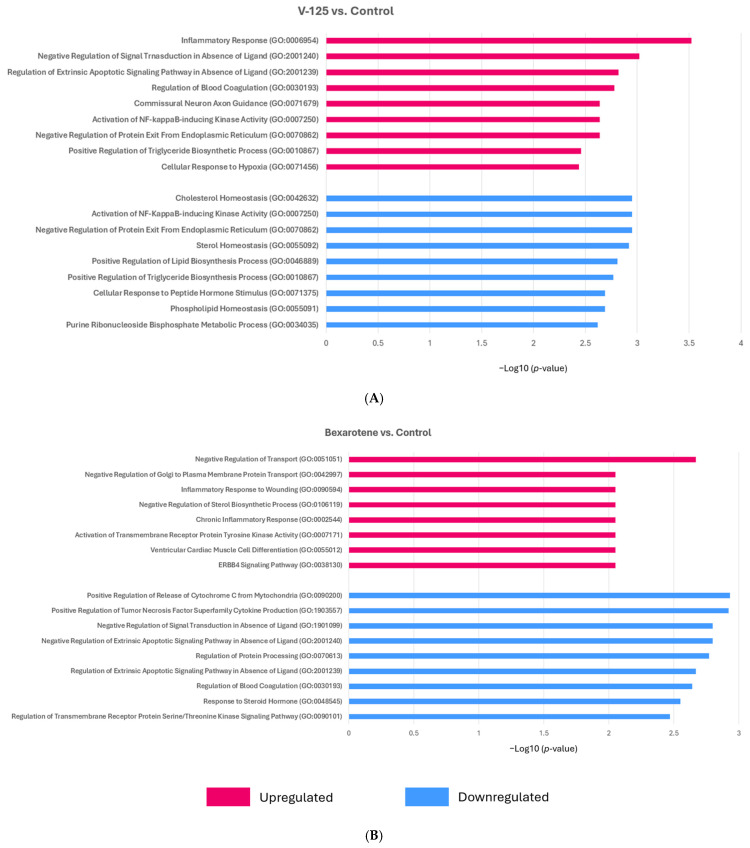
**V-125 and bexarotene enrich distinct pathways in tumors.** Bar graphs display the top differentially enriched pathways identified by GO Biological Process 2025 analysis in Enrichr. (**A**) Upregulated and downregulated pathways altered in V-125-treated vs. control tumors, and (**B**) upregulated and downregulated pathways altered in bexarotene-treated vs. control tumors. Pathway enrichment is plotted as −log10(*p*-value). Pink bars indicate pathways that are upregulated, and blue bars indicate pathways that are downregulated. Pathways are ranked by significance.

**Figure 6 biomedicines-14-00080-f006:**
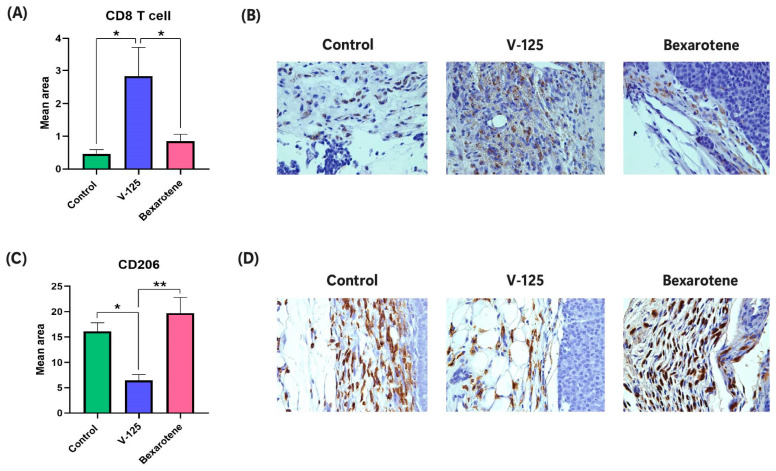
**V-125 modulates immune-related pathways and enhances anti-tumor immune responses in HER2^+^ mammary tumors.** (**A**) Quantification of CD8 cell infiltration (mean positive area) in tumor tissues from control, V-125, and Bexarotene-treated mice. (**B**) Representative IHC images showing CD8 cell staining across treatment groups. For each tumor, regions of interest were manually selected within tumor tissue at 40× magnification for analysis. (**C**) Quantification of CD206^+^ macrophages (mean positive area) in tumor tissues, reflecting tumor promoting macrophage population. (**D**) Representative IHC images showing CD206 staining across treatment groups. For each tumor, regions of interest were manually selected within tumor tissue at 40× magnification for analysis. (**E**) Quantitative RT-PCR validation of *Slit2* mRNA expression in mammary tumors from control, V-125, and bexarotene-treated mice. (**F**) Kaplan–Meier overall survival analysis of breast cancer patients stratified by *SLIT2* expression, showing improved survival associated with higher *SLIT2* levels. Data are presented as mean ± SEM. * *p* < 0.05, ** *p* < 0.01, *** *p* < 0.001, n.s. = not significant; *n* ≥ 3 mice/group.

**Table 1 biomedicines-14-00080-t001:** Primers used for qPCR.

*Nrg1*	(F): 5′-TCCGGCAGAGCCTTCGGTCA-3′
	(R): 5′-TCTCCCGTAGCCTCGGTGGC-3′
*Nfasc*	(F): 5′-TGACCTGGCTGAGAGGAGTGTG-3′
	(R): 5′-AACCTGGAGTGGTCATGCCACA-3′
*Slit2*	(F): 5′-CCATGTAAAAATGATGGCACCTG-3′
	(R): 5′-GTGTTGCGGGGGATATTCCT-3′
*Chi3l1*	(F): 5′-GTACAAGCTGGTCTGCTACT-3′
	(R): 5′-GTTGGAGGCAATCTCGGAAA-3′
*Lrrc26*	(F): 5′-TTGCACTTGGCTGCGTAAGCAC-3′
	(R): 5′-CGGAAAAGCTGTCAGTAGGCTG-3′
*Gapdh*	(F): 5′-CATCACTGCCACCCAGAAGACTG-3′
	(R): 5′-ATGCCAGTGAGCTTCCCGTTCAG-3′
*ABCA1*	(F): 5′-CAGGCTACTACCTGACCTTGGT-3′
	(R): 5′-CTGCTCTGAGAAACACTGTCCTC-3′
*SREBP*	(F): 5′-ACTTCTGGAGGCATCGCAAGCA-3′
	(R): 5′-AGGTTCCAGAGGAGGCTACAAG-3′
*GAPDH*	(F): 5′-GTCTCCTCTGACTTCAACAGCG-3′
	(R): 5′-ACCACCCTGTTGCTGTAGCCAA-3′

## Data Availability

Data is contained within the article or Appendix A. The datasets generated during and analyzed during the current study are available from the corresponding author on reasonable request. For the RNA sequencing, raw data and processed was deposited in the Gene Expression Omnibus and are available through GSE313586.

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
