# Peer review of "RXR Agonist V-125 Induces Distinct Transcriptional and Immunomodulatory Programs in Mammary Tumors of MMTV-Neu Mice Compared to Bexarotene"

_biomedicines, 2025, doi:10.3390/biomedicines14010080_

Round 1

Reviewer 1 Report

Comments and Suggestions for Authors

In their paper, Chowdhury and colleagues investigated the cellular impacts of RXR agonist V-125 in breast cancer tumors and compared it against FDA approved bexarotene. They discovered V-125 induced broader transcriptional reprograming of tumors towards genes associated with improved patient survival when compared to bexarotene. Furthermore, V-125 was demonstrated to promote increase in CD8 cell infiltration and reduction in M2 macrophages in tumor, highlighting the benefit of V-125 application in breast cancer treatment.

The endeavor to improve RXR agonist to treat breast cancer is a relevant subject given the one currently FDA-approved has shown adverse side effects, making research presented in this paper an interesting topic. However, there are several concerns that I have regarding this paper that I believe should be improved or addressed before it is ready to be published:

  1. The authors tried to illustrate that V-125 induced different broader transcriptional program, however in my opinion, it seems that both V-125 and bexarotene induced similar transcription program just at different magnitude level (Figure 1-3).
  2. Do both drugs have similar working concentration as throughout the entire paper same concentration of V-125 and bexarotene were used (600 nM for in vitro and 100 mg/kg for in vivo)? I think a good comparison between drugs can only be made when each drug was used at their most optimal working concentration
  3. In Figure 6, quantification of mean area was used to demonstrate differences in abundance of CD8 T-cell and M2 macrophage infiltration. It would be helpful to illustrate how this quantification was done (by putting circles or arrows for area quantified in representative images).
  4. I would like to see the effect of V-125 and bexarotene in tumor growth. I think this is important to claim benefit of using V-125 over bexarotene for breast cancer tumor treatment.
  5. The quality of the figures is quite poor, particularly figure 2 and 6 (where I struggled to review the representative images)

Author Response

We appreciate the reviewers’ comments to your manuscript. Below are the answers and changes to the manuscript based on the reviewer comments:

  1. The authors tried to illustrate that V-125 induced different broader transcriptional program, however in my opinion, it seems that both V-125 and bexarotene induced similar transcription program just at different magnitude level (Figure 1-3).

We thank the reviewer for this thoughtful comment. We agree that both V-125 and bexarotene modulate some genes in a similar directional trend. However, our interpretation that V-125 induces a broader and more robust transcriptional program is supported by several key observations.

First, V-125 regulates a substantially larger number of genes compared to bexarotene. Specifically, V-125 differentially regulated 71 genes, whereas bexarotene regulated only 17 genes under the same experimental conditions. This difference reflects a markedly broader transcriptional impact unique to V-125.

Second, although bexarotene increased the expression of some genes (e.g., Lrrc26 and Chi3l1), the extent of induction was substantially weaker than that observed with V-125. This difference in magnitude is biologically meaningful, as V-125 consistently produced several fold higher upregulation across multiple targets and showed statistically significant differences compared with bexarotene for all of these genes.

Third, and most importantly, V-125 but not bexarotene significantly induced the expression of clinically relevant genes such as Nrg1 and Nfasc. These genes are associated with improved overall survival in HER2⁺ breast cancer patients based on KMPlot analyses. The absence of induction by bexarotene suggests that these patient-linked transcriptional responses are unique to V-125 rather than simply reflecting a difference in expression magnitude.

Taken together, these findings indicate that V-125 not only induces a greater magnitude of gene expression changes than bexarotene but also preferentially activates genes that are clinically linked to improved patient outcomes.

To address the reviewer’s suggestion, we have also revised the corresponding Result section 3.2 of the manuscript (highlighted in yellow) to more clearly show the direct comparison between V-125 and bexarotene across these target genes.

2. Do both drugs have similar working concentration as throughout the entire paper same concentration of V-125 and bexarotene were used (600 nM for in vitro and 100 mg/kg for in vivo)? I think a good comparison between drugs can only be made when each drug was used at their most optimal working concentration.

We thank the reviewer for raising this important point. In our study, we used the same concentrations of V-125 and bexarotene (600 nM for in vitro assays and 100 mg/kg for in vivo experiments) to allow a controlled, side-by-side comparison of their transcriptional and biological effects. Our aim was to assess how V-125 and bexarotene differ under equivalent exposure conditions. The results demonstrate that even at matched concentrations, V-125 elicits a stronger and broader transcriptional response than bexarotene.

Importantly, previous studies have compared bexarotene with other RXR agonists using similar concentration-matched designs, including both in vitro nanomolar ranges and in vivo dosing around 100 mg/kg (e.g., Lyndsey et al., 2022; Leal et al., 2021; Lyndsey et al., 2023). These studies also demonstrate that comparing RXR ligands at the same concentration provides meaningful insights into their relative transcriptional activities and biological effects.

3. In Figure 6, quantification of mean area was used to demonstrate differences in abundance of CD8 T-cell and M2 macrophage infiltration. It would be helpful to illustrate how this quantification was done (by putting circles or arrows for area quantified in representative images).

We thank the reviewer for this helpful suggestion. To address this comment, we have now uploaded all the representative IHC images in separate PowerPoint files that were used for quantification of IHC images in Figure 6 for full transparency. These images correspond directly to the regions analyzed for CD8⁺ T-cell and CD206⁺ macrophage staining.

We have also expanded and clarified the quantification workflow in Section 2.6 of the Methods (highlighted in yellow). In short, quantification of DAB-positive staining was performed using the Fiji image processing program (ImageJ2). For each tumor, regions of interest (ROIs) were manually selected within tumor tissue at 40× magnification. ROIs were chosen consistently across samples by selecting comparable tumor areas while avoiding tissue folds, or artifacts. DAB signal was quantified using the Color Deconvolution plugin, which separates the brown DAB channel from hematoxylin counterstaining. After deconvolution, the DAB channel image was converted to an 8-bit grayscale image and subjected to a uniform threshold to isolate positive staining. The mean gray value within each ROI was then measured and converted to optical density (OD). For analyses where we report “mean area,” this reflects the area of thresholded DAB-positive signal, providing a relative measure of immune-cell abundance. Multiple ROIs were analyzed per tumor, and results were averaged to generate one value per biological replicate.

4. I would like to see the effect of V-125 and bexarotene in tumor growth. I think this is important to claim benefit of using V-125 over bexarotene for breast cancer tumor treatment.

We appreciate the reviewer’s suggestion to evaluate the effects of V-125 and bexarotene on tumor growth. In response, we have now included tumor volume measurements across multiple time points (Days 0, 3, 7, and 10) for mice treated with control, V-125, or bexarotene. These data demonstrate that V-125 reduces tumor burden more effectively than both the vehicle control and bexarotene over the treatment period.

To support this new analysis, we have added additional details to Method section 2.3. and 2.9. (highlighted in yellow). We also incorporated the corresponding tumor growth results into Section 3.4. (highlighted in yellow). We believe these additions strengthen the manuscript and directly address the reviewer’s concern regarding the comparative impact of V-125 on tumor growth.

5. The quality of the figures is quite poor, particularly figure 2 and 6 (where I struggled to review the representative images)

We appreciate the reviewer’s feedback regarding figure quality. Both Figure 2 and Figure 6 have now been replaced with higher-resolution versions to ensure easier visual interpretation. We believe the updated figures will address the reviewer’s concern.

Reviewer 2 Report

Comments and Suggestions for Authors

The manuscript “RXR Agonist V-125 Induces Distinct Transcriptional and Immunomodulatory Programs in Mammary Tumors of MMTV-Neu Mice Compared to Bexarotene” by Afrin Sultana Chowdhury et al. demonstrates a unique anti-tumor effect of the RXR agonist V-125 by upregulating a distinct transcriptional program, increasing CD8⁺ immune infiltration, and reducing CD206⁺ macrophages in HER2⁺ breast cancer. The findings highlight an interesting role for RXR agonists in regulating tumorigenesis; however, the immunomodulatory aspects of the study require further strengthening.

  1. The RNA-sequencing analysis was performed on bulk tumors, making it unclear whether the differentially expressed genes originate from tumor cells or stromal components. Supplemental Figure 1 suggests that Chi3l1 is expressed in CAFs. The authors should clarify the cellular source of the differentially expressed genes, either by examining JIMT-1 cells directly or by co-staining of the genes of interest with appropriate cell-type markers.
  2. In Figure 4, at least two HER2⁺ cell lines should be tested to rule out the possibility that the observed effects are specific to JIMT-1 cells.
  3. n Section 3.4, both V-125 and Bexarotene induce Abca1 and Srebp expression and inhibit colony formation in JIMT-1 cells. It’s unclear how these in vitro observations relate to tumor growth in vivo. A direct comparison of tumor burden or overall survival in mice treated with V-125 versus Bexarotene is needed to determine whether differences in the tumor immune microenvironment outweigh RXR signaling effects on tumorigenesis.

In Figure 3.6, the IHC images primarily show immune cells located at the tumor border, with minimal infiltration into the tumor core. Images from additional tumor regions should be provided to demonstrate that V-125–treated tumors are truly “hot” tumors with functionally relevant immune infiltration. Moreover, the quantification methods for these images lack sufficient detail. To support the conclusion that V-125 enhances the anti-tumor immune microenvironment, more quantitative analyses—such as CD4/CD8 ratios, T-cell activation markers, and CD8/CD206 ratios assessed by flow cytometry—should be included.

Author Response

We appreciate the reviewers’ comments to your manuscript. Below are the answers and changes to the manuscript based on the reviewer comments:

  1. The RNA-sequencing analysis was performed on bulk tumors, making it unclear whether the differentially expressed genes originate from tumor cells or stromal components. Supplemental Figure 1 suggests that Chi3l1 is expressed in CAFs. The authors should clarify the cellular source of the differentially expressed genes, either by examining JIMT-1 cells directly or by co-staining of the genes of interest with appropriate cell-type markers.

We thank the reviewer for this thoughtful comment. We agree that bulk RNA-sequencing does not allow us to determine whether the differentially expressed genes originate from tumor cells or stromal/immune compartments. However, identifying the precise cellular source of each transcript was not within the scope of the current study, which aimed to evaluate the overall transcriptional and immunomodulatory effects of V-125 compared with bexarotene in the tumor microenvironment. We fully acknowledge that single-cell RNA-sequencing or spatial transcriptomics would be the most appropriate approaches to resolve this question, and we plan to explore these strategies in future work.

Previously, we performed conditioned-media experiments using supernatant from the MMTV-neu–derived E18-14C-27 cancer cell line on bone-marrow-derived macrophages. These preliminary assays showed that V-125 significantly induced the expression of Chi3l1, Nrg1, and Nfasc in macrophages, whereas bexarotene induced markedly weaker responses. While these data do not fully delineate the cellular origin of these transcripts in vivo, they suggest that at least some of the observed gene expression changes may arise from the immune compartment. These data are intended for future publication, and while these data are not being included in the manuscript, we are providing them here for the reviewer.

We also agree with the reviewer that determining whether these genes are derived from cancer cells, immune cells, or other stromal populations will be an important next step, and we intend to investigate this more directly in future studies.

2. In Figure 4, at least two HER2⁺ cell lines should be tested to rule out the possibility that the observed effects are specific to JIMT-1 cells.

We thank the reviewer for this important comment. We agree that including more than one HER2⁺ cell line can strengthen the interpretation of our findings. To address this, we evaluated the E18-14C-27 cell line derived from the MMTV-neu mouse model of HER2⁺ breast cancer.

We conducted colony formation assays in this murine cell line. However, neither V-125 nor bexarotene produced measurable inhibition of clonogenic growth in E18-14C-27 cells. These results suggest that the functional response to RXR agonists may be cell line specific, likely reflecting intrinsic biological differences between HER2⁺ models. Notably, JIMT-1 cells are known to be stem-cell like and trastuzumab-resistant, features that may contribute to their heightened sensitivity to RXR-targeting compounds compared with the more differentiated MMTV-neu derived E18-14C-27 cells.

Importantly, the absence of a direct inhibitory effect of V-125 in E18-14C-27 cells strengthens our overall conclusion that V-125’s anti-tumor activity in vivo is largely immune-mediated rather than driven solely by tumor-intrinsic growth suppression. The differential response between the two HER2⁺ models aligns with the hypothesis that V-125 exerts its efficacy primarily by modulating the tumor immune microenvironment.

Although we were unable to include additional human HER2⁺ cell lines due to time constraints, we have added the E18-14C-27 colony-formation results to the supplemental material to demonstrate that we explored this point experimentally. We have also updated the manuscript text to reflect the inclusion of this second HER2⁺ model.

3. In Section 3.4, both V-125 and Bexarotene induce Abca1 and Srebp expression and inhibit colony formation in JIMT-1 cells. It’s unclear how these in vitro observations relate to tumor growth in vivo. A direct comparison of tumor burden or overall survival in mice treated with V-125 versus Bexarotene is needed to determine whether differences in the tumor immune microenvironment outweigh RXR signaling effects on tumorigenesis.

We appreciate the reviewer’s thoughtful comment. In Section 3.4., our goal in examining Abca1 and Srebpexpression in JIMT-1 cells was to validate RXR target engagement for both V-125 and bexarotene. These genes were selected because they are well-established RXR-responsive targets. As noted, both compounds significantly upregulated Abca1 and Srebp mRNA levels compared to vehicle-treated controls, confirming that each agonist effectively activates RXR-dependent transcriptional programs in vitro.

To address the reviewer’s concern regarding the relationship between these in vitro findings and in vivo tumor outcomes, we have now included a new figure comparing tumor volume among MMTV-neu mice treated with V-125, bexarotene, or vehicle control. This analysis shows that V-125 reduced tumor burden compared to both the control and bexarotene groups, despite similar RXR activation in vitro. These in vivo differences align with our transcriptional and immunophenotypic analysis and support the interpretation that V-125’s enhanced antitumor effects may be driven by its ability to modulate the tumor immune microenvironment. We have updated Figure 4 and Results section 3.4 (highlighted in yellow) to include this newly added tumor-volume comparison. We also have added additional details to Method section 2.3. and 2.9. (highlighted in yellow). We believe these additions strengthen the manuscript and directly address the reviewer’s concern regarding the comparative impact of V-125 and bexarotene on tumor burden.

4. In Figure 3.6, the IHC images primarily show immune cells located at the tumor border, with minimal infiltration into the tumor core. Images from additional tumor regions should be provided to demonstrate that V-125–treated tumors are truly “hot” tumors with functionally relevant immune infiltration. Moreover, the quantification methods for these images lack sufficient detail. To support the conclusion that V-125 enhances the anti-tumor immune microenvironment, more quantitative analyses—such as CD4/CD8 ratios, T-cell activation markers, and CD8/CD206 ratios assessed by flow cytometry—should be included.

We thank the reviewer for these insightful suggestions. In response, we have uploaded the representative raw IHC images used for quantifying CD206⁺ macrophages and CD8⁺ T cells in Figure 6. These unedited images correspond directly to the tumor regions included in our analysis and provide full transparency into the areas evaluated.

Regarding immune cell localization, it is important to note that our IHC samples were collected at an early stage of tumor progression, during which immune cells frequently accumulate at the tumor margin rather than uniformly infiltrating the tumor core. This peripheral localization pattern is well-documented in the MMTV-neu model, particularly in early and mid-stage lesions. Nonetheless, the raw images now provided include additional tumor fields to better illustrate the distribution of immune cells across the tissue.

We have also expanded the Methods section 2.6 (highlighted in yellow) to describe our quantification workflow in detail, including the use of Fiji/ImageJ color deconvolution, ROI selection, and optical density measurement procedures.

We acknowledge the reviewer’s suggestion to include more extensive immunophenotyping such as CD4/CD8 ratios, T-cell activation markers, and CD8/CD206 ratios by flow cytometry. Due to practical constraints, including the need to enroll additional mice, wait for tumors to reach ~5×5 mm, and then conduct treatment and analysis, we were unable to perform a full set of flow cytometry experiments in the revision timeframe.

However, we did perform preliminary flow cytometric analysis in two mice prior to the start of revision activities, assessing the CD8/CD4 ratio in V-125–treated versus control tumors. These preliminary data are included as a new supplemental figure for the reviewer’s reference. While limited by sample size, the results are consistent with our IHC findings and support the conclusion that V-125 promotes a more pro-inflammatory tumor immune environment.

We fully agree that deeper immune profiling, including activation markers and macrophage/T-cell ratios will be important, and we plan to pursue these analyses in future studies as part of a more comprehensive characterization of V-125 mediated immunomodulation.

Round 2

Reviewer 2 Report

Comments and Suggestions for Authors

The revised manuscript has fully addressed all the issues, and I recommend accepting it. 

Author Response

We thank the reviewer for the comments and reviews to the manuscript